# Towards a Machine Learning Model for Cognitive Performance Prediction

Laura Ginestretti, Jacopo Lazzari, Alessandro Verosimile, Susanna Bardini, Marco D. Santambrogio

Politecnico di Milano, Dipartimento di Elettronica Informazione e Bioingegneria (DEIB), Milan, Italy

{laura.ginestretti, alessandro.verosimile, susanna.bardini, marco.santambrogio}@polimi.it

jacopo.lazzari@mail.polimi.it

*Abstract*—**Predicting cognitive performance by leveraging physiological, lifestyle, and demographic data is vital for advancing personalized mental health management and improving individual outcomes. This study presents an explainable regression framework designed to predict cognitive performance while providing meaningful and interpretable insights aimed at its improvement. To capture both nonlinear and individual-specific relationships present in multimodal datasets, we utilize machine learning models such as XGBoost and Linear Generalized Additive Models (GAM). In this scenario, interpretability plays a central role in translating machine learning model predictions into actionable knowledge for cognitive performance improvement. To this end, we employ SHapley Additive Explanations (SHAP) for global and local feature attribution, which reveals how factors such as physical activity, respiration patterns, and heart rate variability impact cognitive outcomes at individual level. Moreover, Diverse Counterfactual Explanations (DiCE) are used to generate multiple plausible and realistic modifications to input features, giving the possibility to elaborate personalized recommendations that can improve predicted cognitive performance. The proposed framework is validated on two distinct datasets: the first one comprising real-life oral presentations evaluated by external experts, where XGBoost achieved a 42.5% improvement over baseline; the second one based on standardized computerized cognitive assessments, where Linear GAMs reached a 46.8% improvement. These results demonstrate the benefits of combining strong predictive accuracy with interpretability, supporting tailored cognitive health monitoring and intervention planning. Overall, this work highlights the benefits of multifactorial, explainable machine learning models to optimize individual cognitive performance through personalized, data-driven strategies.**

*Index Terms*—**Cognitive Performance Prediction; Explainable Machine Learning; Personalized Health Monitoring; Counterfactual Analysis**

## I. INTRODUCTION

Cognitive function and performance are fundamental components of overall well-being, influencing not only **academic and professional success** but also **daily practices** such as decision-making, social interactions, and adaptability. From a clinical perspective, Cognitive Performance (CP) is an essential indicator of neurological and mental health, often used to assess cognitive decline, track the progression of conditions like dementia, Alzheimer's, and other cognitive disorders, and guide therapeutic interventions [1]. Monitoring and improving CP can thus be critical not only for **optimizing daily functioning** but also for **preventing or managing cognitive impairments** associated with aging or

neurological conditions. In this scenario, just as the brain is a highly complex organ, continuously shaped by both internal physiological processes and external lifestyle influences, our CP is likewise the result of a dynamic interaction between intrinsic factors, such as physiological condition and genetic predisposition, and extrinsic factors, including environmental context and daily habits [2]. As a result, recent research has focused on exploring the relationship between CP and these key lifestyle elements, including sleep [3], Physical Activity (PA) [4], nutrition [5] and various physiological parameters. However, while works present in the literature have been establishing general correlations between individual lifestyle factors and CP, **no comprehensive Machine Learning (ML) framework** currently **integrates all these aspects simultaneously** to predict CP [6], [7]. This is indeed an essential element to be taken into account, given the high complexity of the interactions between the factors themselves. Moreover, existing literature lacks a framework capable of making **CP predictions while accounting for individual baseline** cognitive abilities. Ultimately, no current approach provides a systematic method to **identify personalized adjustments** in lifestyle and physiological parameters aimed at CP enhancement. Therefore, the **objectives of this study** are threefold: first, to develop an **extensive analysis** of the **influence of lifestyle and physiological factors on CP**, considering their combined impact rather than isolated correlations; second, to design a **ML-based predictive framework** that incorporates both generalizable insights and individual-specific considerations, accounting for inherent cognitive abilities; third, to present **personalized adjustments** tailored to individuals, identifying specific lifestyle and physiological modifications that can effectively enhance CP.

This paper is structured as follows: Section I reviews the state of the art, while Section II and Section III outline the data collection protocol and describe the predictive frameworks employed in the study for CP prediction. Section IV reports the experimental findings, which are further analyzed and discussed in Section V. In section Section VI we draw our conclusions and propose future works.

### A. Experimental Studies on CP Correlations with Lifestyle factors

The influence of **PA** on CP has been extensively studied, demonstrating both immediate and long-term effects on brain

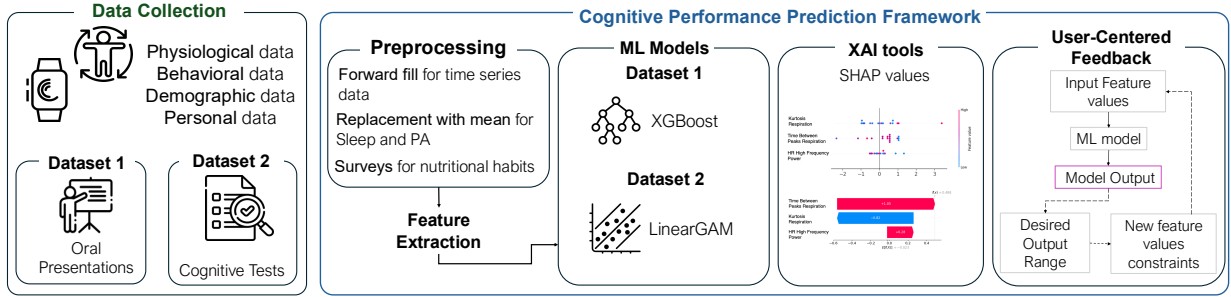

Fig. 1. The proposed framework preprocesses and standardizes physiological, behavioral, demographic, and personal data. Features are selected and transformed to train ML models, with performance evaluated and interpreted through explainability tools to understand how feature variations affect CP scores.

function, mental health, and overall cognitive well-being [8]. For instance, a single session of moderate-intensity exercise has been shown to acutely **enhance cognitive functions** [9]. This happens primarily due to increased cerebral blood flow and elevated neurotransmitter levels, such as dopamine, which plays a crucial role in cognitive processes. Similarly, **sleep quality and duration** play a critical role in optimizing mental functions [10]. Research indicates that individuals who consistently achieve adequate sleep duration **perform better on cognitive tasks**, including memory recall and executive function assessments, compared to those with insufficient sleep. Finally, **nutrition** is a key factor in CP, **influencing mental functions** through both immediate and long-term mechanisms. Importantly, the **interaction between macronutrients** has been demonstrated to be particularly relevant for CP. Studies analyzing carbohydrate-to-protein ratios indicate that balanced meals (e.g., a 1:1 ratio) can enhance cognitive functions such as reaction time and decision-making more effectively than meals dominated by a single macronutrient [11].

### B. ML Frameworks for CP Prediction

In this scenario, several studies have leveraged ML techniques to predict CP starting from physiological signals. For instance, Empatica wearable sensors have been utilized to collect Heart Rate (HR), temperature, and skin conductance data while participants engaged in cognitive tasks [12]. Among those, one demonstrated the potential of these physiological signals in predicting CP during academic exams [12]. Moreover, Empatica E4 wrist-worn have been also exploited in study involving individuals with mild cognitive impairment to collect physiological data, then used to predict neuropsychological test scores. This demonstrated the potential of wearable technology for continuous cognitive monitoring and early detection of cognitive decline, underscoring the feasibility of wearable-based CP prediction in real-world settings. Building on this foundation, recent advances in ML-based CP predictions have incorporated Explainable AI methods, such as **SHapley Additive Explanations (SHAP)** [13], **Integrated Gradients** [14], [15], and **Occlusion Sensitivity** [16], to explain how specific features contribute to CP predictions, enhancing model transparency and interpretability.

## II. DATASET & PREPROCESSING

Since, to the best of our knowledge, no previous work in the literature simultaneously analyzes different lifestyle factors and physiological data, together with demographic and personal information, it was necessary to **collect new data** for this study. The data collection process was structured around two different experimental paradigms, each designed to explore CP through a distinct modality. The first dataset focused on real-life performance in oral presentations, while the second employed standardized computerized tasks. This dual approach allowed us to investigate CP under different conditions, capturing both everyday social contexts and individually administered testing environments. All participants were healthy, with no history of neurological, psychiatric, or cardiometabolic conditions, and gave written informed consent. Only non-invasive data were collected, with no clinical interventions or sensitive information required, in full compliance with ethical standards and participant privacy. In the following sections, the collected datasets are presented, along with the bias removal techniques applied and the preprocessing steps undertaken to prepare the data for analysis.

### A. Dataset 1

The first data collection involved 30 participants (age: 23.43 ± 7.36, 21 males, 9 females). **Physiological parameters, sleep, and PA** were recorded using the *Garmin vivoactive 5* smartwatch [17]. **Nutritional data** were logged via a dedicated mobile app [18], capturing meal composition and macronutrient intake. Additionally, **metabolic and personal parameters** (including age, sex, height, weight) were collected through surveys.

CP was assessed through evaluations of **participants' ability to deliver oral presentations**. This task was chosen because it represents a real-life CP scenario, and enables us to analyze correlations **in practical, everyday contexts**. The final dataset comprised 84 oral presentations distributed across all participants. Each evaluator had no prior relationship with the participants, in order to minimize potential bias arising from personal familiarity. Evaluation bias is crucial in this study, as external assessors often introduce subjectivity [19]. Without bias correction, evaluations reflect assessors' biases, confounding factors, and participants' baseline cognitive skills.

To address this, we adopted a Bayesian framework based on Variational Inference (VI) [20], applied to a model introduced in a previous work [19], to ensure a more reliable and unbiased assessment of each subject's performance. Specifically, the observed score assigned by evaluator $v$ to work $u$, denoted as $y_{uv}$, is modeled as: $y_{uv} \sim \mathcal{N}(s_u + b_v, \frac{1}{\tau_v})$, where $s_u$ represents the true score of the work, $b_v$ denotes the bias introduced by evaluator $v$, and $\tau_v$ captures the reliability of reviewer $v$. We define prior distributions for the latent variables $\theta = \{s_u, b_v, \tau_v\}$ and use VI to optimize the Evidence Lower Bound (ELBO), approximating their posterior distributions. The approximate posterior of $s_u$ is then used to sample adjusted true scores, compensating for evaluator bias and reliability. At the conclusion of this process, participants' scores are rescaled as shifts relative to their own mean vote.

### B. Dataset 2

In this second data collection, physiological parameters, sleep, and PA were recorded using the *Garmin vivoactive 5* smartwatch. Additionally, personal parameters, including age, sex, height, weight, were collected. The dataset comprised 14 participants (age: $31.86 \pm 10.45$, 11 males, 9 females).
CP was assessed through the overall score provided by the Cambridge Brain Sciences Cognitive Assessment [21]. This assessment consists of a series of **standardized tasks** designed to measure core cognitive domains such as memory, attention, reasoning, and problem-solving. Each participant to the study took the test for a total of 12 times over a 6 weeks period.
While less reflective of real-world social contexts, these tasks enable precise and replicable assessment of specific cognitive functions. To limit the effects of the learning curve on the exercises proposed , which are limited in number and therefore repeated over time, participants' final scores were rescaled to reflect their deviation from their average performance, calculated within a **moving window** of 5 attempts, including the 2 previous and 2 subsequent trials. This approach ensures that each participant's performance is evaluated relative to their own baseline over time, removing confounding effects due to the increasing familiarity with the tasks.

### C. Data Preprocessing

In this study, imputation techniques for missing values within the dataset have been applied. For **time-series data**, missing values were handled using **forward-fill**. This method was chosen as it respects the natural causal flow of time by using past information to fill gaps, rather than relying on future data points, which may not be available or realistic in many scenarios [22]. Imputation of time series data was performed on 20.8% of the series for Dataset 1 and on 37.1% for Dataset 2. For Dataset 1, 60% of the time series had less than 25% missing data, while for Dataset 2, 50% had less than 25%. In both datasets, no series had more than 75% missing data.

Moreover, scalar data, such as **Sleep and PA**, were imputed through the average of the participant's collected data.
No missing values were observed in **demographic and metabolic data**, as these were collected at the registration of the participants in the study. For **nutritional data**, missing values caused by incomplete logs were imputed using the participant's mean intake for the same meal type on other days, ensuring that imputations reflected realistic dietary patterns.

### III. PROPOSED METHODOLOGY

This section describes our proposed methodology, where ML models predict CP from physiological, demographic, behavioral, and personal data. Explainability is provided via SHAP values for feature importance, while Diverse Counterfactual Explanations (DiCE) assess how input changes impact predictions, supporting feedback and interventions.

### A. Feature Extraction

The **feature extraction** process in this study was designed to transform raw time-series data extracted from smartwatch into meaningful inputs for ML models. The key data sources for this process included physiological signals, specifically HR, respiration rate, and stress levels. For the time-series data, different kinds of features have been extracted. First, basic **statistical metrics** such as mean, standard deviation, and variance, were extracted to capture the central tendency and variability of the time-series data. Second, the **zero-crossing rate** was calculated, which measures how frequently the signal crosses the zero value, providing insight into the frequency and dynamics of the physiological signal. **Peak characteristics**, including the height and timing of peaks within the signal, have been calculated to capture sudden changes in physiological parameters. Last, **frequency-domain features** including statistics of the power spectrum, dominant frequency, spectral centroid, and spectral flatness, were extracted to capture key frequency characteristics of the signal.

These extracted features constituted the dataset along with non-time-series variables such as demographic information and personal data. The final dataset therefore included **106 standardized features** categorized as follows:

- **Extracted Features**: 26 features per time-series (HR, respiration, stress), totaling 78.
- **Static Features**: 10 demographic and anthropometric attributes (age, gender, BMI, height, weight).
- **Physical Activity and Sleep Features**: 6 features summarizing activity and sleep data.
- **Nutritional Features**: Aggregated dietary logs capturing macronutrient intake per meal.

### B. Model Selection and Optimization

In this study, various ML models were explored to predict CP based on the extracted features. The models considered **Random Forest (RF)**, **XGBoost** and **Linear Generalized Additive Model (LinearGAM)**, as these are known for their effectiveness in handling complex datasets and providing high predictive accuracy. Model performance was evaluated using the Leave-One-Out Cross-Validation method, ensuring robust performance assessment. Since no existing models in the literature address the task of CP prediction starting from physiological, demographical, personal and behavioral data,

to evaluate these models we established a **random guess baseline** for reference. Therefore, we included a **naive model that always predicts zero as the target value**. In the context of a bias-corrected framework, this constant prediction corresponds indeed to the individual's average performance, meaning the model does not extract any additional information from the input data and consistently predicts no deviation from this average.

In this study, feature selection techniques such as forward, backward, and bidirectional selection were used to enhance ML model performance by identifying the most relevant features for predicting CP, reducing model complexity while preserving key variables [23].

To optimize these models, **hyperparameter tuning** was performed using Optuna [24], an automatic hyperparameter optimization framework. The optimized hyperparameters were selected based on the model's performance, using the Mean Absolute Error (MAE) as evaluation metric.

### C. Explainability and Counterfactuals Generation

To provide explainability tools for understanding the underlying patterns in the model's predictions, **SHAP** [25] was employed. This method was applied after the model's predictions on the test set, offering insights into the decision-making process of the models and highlighting the contribution of each individual input to the final predictions. To determine how much a single feature $i$ contributes to a specific output, SHAP evaluates all possible subsets of features that exclude $i$ and calculates the model's prediction for each subset. Then, feature $i$ is added back to these subsets to observe **how much the prediction changes**. The change in the prediction represents the contribution of feature $i$ in the context of that specific subset. By assigning importance scores to each feature, SHAP allowed us to assess how physiological, nutritional, or demographic data impacted CP predictions. This analysis helped to **identify the key drivers of the model's behavior**, offering interpretability and actionable insights into the relationships between input features and target variables.

While feature attribution methods like SHAP provide insights into which features drive a model's predictions, they do not offer guidance on how to alter inputs to achieve a desired outcome. In this context, **counterfactual explanations** provide an alternative approach to interpretability, focusing on identifying changes in input features that would lead to a different prediction [26]. Among various counterfactual generation methods, **DiCE** has emerged as a robust framework for producing multiple plausible counterfactual instances [27]. In particular, given an instance $x$ with model prediction $\hat{y}$, the goal of counterfactual explanations is to find a counterfactual $x'$ such that $f(x') = \hat{y}_{cf}$, where $\hat{y}_{cf} \neq f(x)$, where $f(x)$ represents the original model prediction, $x'$ is the modified input instance, and $\hat{y}_{cf}$ is the desired target prediction. To ensure realistic counterfactuals, feature changes are limited to the minimum and maximum values observed in the training set, preventing out-of-distribution modifications and keeping explanations within valid data bounds.

TABLE I
MODEL PERFORMANCES COMPARED TO RANDOM GUESS BASELINE

| | Model | MAE | Improvement (%) |
|---|---|---|---|
| **Dataset 1** | **XGBoost** | **0.1545** | **42.5%** |
| | Random Forest | 0.1863 | 30.6% |
| | LinearGAM | 0.2007 | 25.28% |
| | Random Guess Baseline | 0.2686 | 0.0% |
| **Dataset 2** | **LinearGAM** | 0.8277 | 46.8% |
| | XGBoost | 1.2165 | 21.8% |
| | Random Forest | 1.2997 | 16.4% |
| | Random Guess Baseline | 1.5554 | 0.0% |

### IV. RESULTS

This section presents the results of the proposed CP prediction framework, both in terms of performance and explainability. Both SHAP and DiCE results will be demonstrated on a few selected instances from the test sets of Dataset 1 and Dataset 2, as each result is inherently unique.

### A. Model Performance

As regards performance on Dataset 1, **XGBoost** demonstrated the best predictive performance among the evaluated models, achieving a **MAE of 0.1545** and an **improvement of 42.5%** over baseline predictions (Table I).

In our second dataset, among the evaluated models, **LinearGAM** demonstrated the best predictive performance, achieving a **MAE of 0.8277** and an **improvement of 46.8%** over baseline predictions (Table I).

### B. Feature Importance and Instance-Specific Analysis via SHAP

In both Fig. 2 and Fig. 3, the left side displays the global SHAP summary plot for the test features, while the right side shows the SHAP values calculated for an individual instance. On the left, the image displays a SHAP beeswarm plot illustrating the **impact of selected features on the model's output**. Each point represents a SHAP value for a specific feature and individual instance, showing how much that feature shifts the prediction away from the baseline. For every instance, a point indicates the feature's value, colored according to a gradient from blue to red, where blue represents low values and red represents high values. For example, as seen for HR_Energy in Fig. 2, higher feature values contribute positively to the prediction, appearing on the right side of the zero line, while lower values push the prediction negatively. On the right, the graph displays the selected features along the y-axis, each accompanied by a bar whose length indicates the magnitude of that feature's contribution to the final CP prediction. The color of the bars reflects the direction of the effect: red bars signify positive contributions (increasing CP), while blue bars indicate negative contributions (decreasing CP). The value $f(x)$ represents the model's final predicted output, highlighting the features that have the strongest influence on the result. For instance, the SHAP graph on the right of Fig. 2 illustrates the impact of features for a specific individual from Dataset 1. In this case, high PA minutes had

TABLE II
COUNTERFACTUAL FEATURE MODIFICATIONS FROM DATASET 1.

| Feature | Original | Counterfactual |
| --- | --- | --- |
| PA Minutes | 10.0 | 95.0 (↑) |
| HR Freq. Std | 0.15 | 0.32 (↑) |
| Resp. Quant 25 | 4.2 | 4.2 |
| HR Max. Frequency | 0.62 | 0.62 |
| **CP** | -0.15 | 0.08 |

TABLE III
COUNTERFACTUAL FEATURE MODIFICATIONS FROM DATASET 2.

| Feature | Original | Counterfactual |
| --- | --- | --- |
| Time Between Peaks Respiration | 3.67 | 3.67 |
| Kurtosis Respiration | 0.53 | 0.62 (↑) |
| HR High-Frequency Power | 431.79 | 431.79 |
| **CP** | 0.06 | 0.21 |

a strong positive contribution (+0.46) to the CP prediction, clearly indicated by the red bar next to `PA_Minutes`, which is the longest among all features.

### C. DiCE Counterfactual Analysis

To complement SHAP-based analysis, counterfactual explanations were applied to the XGBoost and LinearGAM models, focusing specifically on test set instances. Rather than just improving interpretability, this approach provides **actionable suggestions** by identifying minimal feature changes that could improve predicted CP. A representative example is provided in Table II demonstrating how counterfactuals can **shift a negative CP prediction to a positive one** by suggesting realistic modifications to input features. The original feature values are shown alongside their modified counterparts after applying the counterfactual intervention. `PA_Minutes` increased significantly from 10.0 to 95.0, and `HR_Frequency_Standard_Deviation` rose from 0.15 to 0.32. `Respiration_Quantile_25` and `HR_Maximum_Frequency` remained unchanged at 4.2 and 0.62, respectively.

Table III shows the counterfactual feature modifications for an instance from Dataset 2. The original model prediction was -0.06, which shifted to 0.21 after applying the counterfactual. The table reports the feature values before and after the counterfactual modification. Specifically, `Time_Between_Peaks_Respiration` remained unchanged at 3.67 seconds, the `HR_High-Frequency_Power` stayed constant at 431.79, whereas the `Kurtosis_Respiration` increased from 0.53 to 0.62.

## V. DISCUSSION

In this section, we examine both the predictive performance of the models and the extent to which the most influential features align with biologically meaningful patterns. Indeed, using SHAP and DiCE analyses, we uncovered physiological factors influencing CP and emphasized the value of personalized insights for targeted interventions.

### A. Model Performance

As highlighted in Table I, XGBoost emerged as the best-performing model for **Dataset 1**, selecting features such as `PA_Minutes`, `Resp_25th_Quantile`, `HR_Freq_Std`, `HR_Energy`, and `HR_Max_Features`. The superior performance of XGBoost can be attributed to its ability to **effectively handle non-parametric and potentially nonlinear relationships within the data**, which traditional parametric models might struggle to capture. This flexibility allows XGBoost to model complex interactions and intrinsic patterns that are present in physiological and lifestyle-related features. Furthermore, an examination of the selected features reveals that the task could have been heavily influenced by different stressors. This explains the significant role that PA plays in the model, as exercise is well-known to mitigate stress and reduce nervousness, thereby impacting CP.

As shown in Table I, the LinearGAM was the model achieving the best performance on Dataset 2. This result suggests that **the relationships in this dataset are well captured by additive models with smooth terms**. Indeed, these models are particularly effective when the data exhibit structured yet potentially nonlinear trends, while maintaining high interpretability. This confirms **relationships in Dataset 2 are likely more regular and additive in nature**, with fewer non-linear interactions between features. In terms of selected features, Dataset 2 focuses on `Time_Between_Peaks_Respiration`, `Kurtosis_Respiration`, `HR_High_Frequency_Power`; their inclusion suggests that the task involved emphasizes more refined and stable physiological signals.

The comparison between the two datasets highlights how the **optimal model choice strongly depends on the underlying data characteristics**. In Dataset 1, which appears to contain complex, nonlinear relationships among physiological and behavioral features, XGBoost outperformed other models due to its ability to model such intricacies. Conversely, in Dataset 2, where the data follows more regular, additive patterns, LinearGAM achieved the best performance while also offering enhanced interpretability. These findings underscore the importance of **tailoring model selection to the specific structure and nature of the dataset** at hand.

### B. Feature Importance and Instance-Specific Analysis

Starting from the beeswarm plot belonging to Dataset 1 (left graph of Fig. 2), we observe that features exhibit **highly individualized effects on CP**. The exception in this context is the `Previous_Day_PA`: higher levels of PA on the preceding day are consistently associated with improved CP, which aligns with existing evidence highlighting the **benefits of moderate exercise for neuroplasticity and executive functioning**. Another notable example is `HR_Energy`: positive `HR_Energy` values correlate with enhanced CP. `HR_Energy`, indeed, quantifies the overall power of HR fluctuations, implying that higher values reflect sustained physiological activation correlating with improved focus, information processing, and

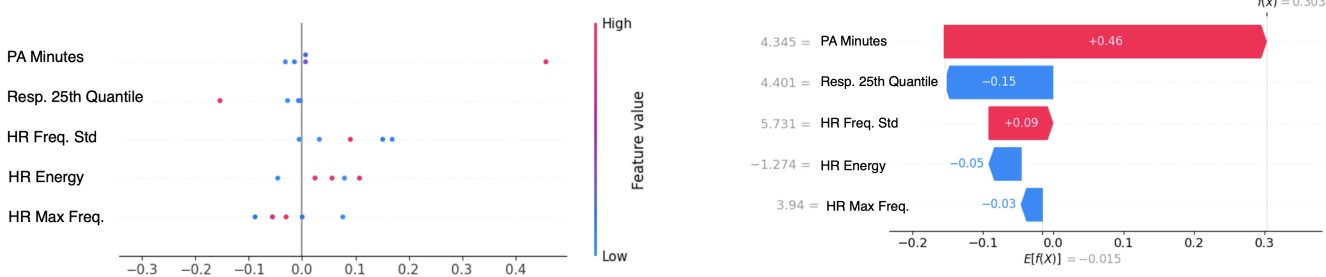

Fig. 2. SHAP plots for Dataset 1: the beeswarm plot (left) shows feature impact on model output, with red (high) and blue (low) feature values along the x-axis of SHAP values. The waterfall plot (right) visualizes feature contributions for one test instance; bar length indicates impact, with red for positive and blue for negative contributions.

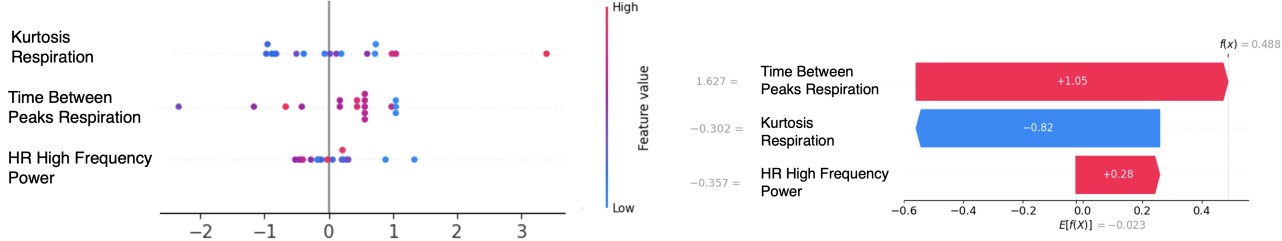

Fig. 3. SHAP beeswarm and waterfall plots for Dataset 2.

overall cognitive efficiency.

As regards the right-side SHAP graph from Fig. 2, it illustrates the impact of features on the specific individual of Dataset 1. In this case, high PA minutes had a positive contribution (+0.46), while elevated respiration quantile values negatively impacted the CP prediction (-0.15). On the other hand, HR variability was positively correlated with CP (+0.09).

SHAP analysis from Dataset 2 reveals once again that the direction and magnitude of influence of the features on the model's prediction vary across individuals. Among these, Kurtosis_Respiration displays a wide range of SHAP values, despite its feature values remain predominantly low across the test set. This suggests a weak or non-linear relationship between respiratory irregularity and CP. Such pattern is consistent with the idea that the cognitive effects of stress or dysregulated breathing patterns may depend on individual baseline stats, potentially impairing CP or reflecting cognitive systems activation. In contrast, Time_Between_Peaks_Respiration shows a more consistent and predominantly positive contribution to CP prediction, with higher features values associated with higher SHAP values. This suggests a clearer relationship, where longer intervals between respiratory peaks may be indicative of favorable physiological conditions for cognitive performance. Meanwhile, HR_HighFrequency_Power exerts a modest influence on the model's output, with less variation across individuals when compared to the other features. This may imply that its predictive value emerges only in specific physiological contexts or through interactions with other features, rather than acting as a strong standalone predictor. Findings from both

datasets provide insight into the physiological mechanisms underlying CP, but a deeper understanding of the model's decision-making requires analysis of SHAP values at the individual level. While **global SHAP patterns reveal general trends**, it is the **individual physiological baselines** that shape the influence of features on **localized predictions**, highlighting the importance of personalized approaches to CP assessment.

### C. DiCE Counterfactual Analysis

The generated counterfactuals for Dataset 1 highlight the crucial role of PA in improving CP predictions. Specifically, achieving an improved CP prediction requires a significant increase in PA minutes, as demonstrated in Table II. This underscores the importance of **sustained and substantial physical activity in enhancing CP outcomes**. Additionally, the counterfactual indicates an increase in HR frequency standard deviation (HR_Freq_Std) from 0.15 to 0.32. This change is supported by the idea that higher PA levels, especially when performed with intensity, can positively influence heart rate metrics. The other features, Respiration_Quantile_25 and HR_ Maximum_Frequency, remain unchanged in this example, implying that their impact on the CP prediction is less significant within this context.

As regards Dataset 2 instance, reported in Table III, only one feature required adjustment to significantly increase the predicted CP: the Kurtosis_Respiration, which shifted from 0.53 to 0.62. This change illustrates how **small variations in respiratory distribution**, potentially related to stress or irregular breathing patterns, **can influence CP predictions at the individual level**. Meanwhile, key features such as Time_Between_Peaks_Respiration and

`HR_High-Frequency_Power` remained unchanged, suggesting that these variables may serve as stable physiological baselines with limited leverage for change in certain contexts. While these observations are drawn from a specific instance, a similar pattern emerges across multiple cases, although there is variability in which features are most influential. Overall, these counterfactual insights emphasize the **need to focus on actionable and feasible interventions** that can realistically influence the features driving CP improvements. Therefore, careful selection of modifiable features is crucial to ensure practical and effective recommendations to be generated.

## VI. Conclusions

In this study, we have presented a **comprehensive ML framework for predicting CP** by integrating a wide range of physiological, personal, behavioral and demographic factors. Our approach leverages models, such as XGBoost and LinearGAM, demonstrating their effectiveness in capturing complex and individual-specific relationships within heterogeneous datasets. The application of SHAP values provided valuable **insights into the contribution of distinct features at both the global and individual levels**, highlighting the pivotal roles of physical activity, respiratory patterns, and heart rate variability in shaping CP. Moreover, the incorporation of DiCE offered a **practical way for personalized interventions** to be generated, enabling the identification of feasible lifestyle adjustments that can potentially enhance cognitive outcomes. Overall, our findings emphasize the **importance of adopting a multifactorial and personalized perspective** when modeling CP, moving beyond isolated factor analyses towards integrated predictive and explainable frameworks.

Future work will focus on expanding the dataset size and diversity, incorporating longitudinal data to capture temporal dynamics, and exploring causal inference methods to strengthen the interpretability and actionability of the model recommendations. Additionally, integrating real-time feedback mechanisms could further support individualized cognitive health management in everyday settings.

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
