# OpenReview forum: "Towards a Machine Learning Model for Cognitive Performance Prediction"
_IEEE.org/EMBS/BHI/2025/Conference — BHI 2025_

### Official Review · Reviewer_CC3Y · 2025-07-05
**Revision 1**

**Confidence:** 3
**Clarity Of Writing:** good
**Clinical Significance:** good
**Methodological Novelty:** great
**Overall Rating:** 7

**Experiments And Results:**

great

**Questions For The Authors:**

As pointed-out for the paper weaknesses:

1) Ethical reference approval for the current study is missing in the manuscript

2) The authors did not disclose the percentage of missing physiological data obtained from the smartwatch comparative to the total amount of data collected, which will allow readers to get an estimate of the imputation work required to complete the dataset.

**Strengths:**

1) Overall it is a well-written manuscript, with authors providing enough details for the proposed method(s).

2) Use of a significant number of trialed individuals.

3) Large number of extracted features, ranging from physiological variables to demographic and personal data.

4) The use of SHAP and DiCE metrics to evaluate the contribution and adjustment of distinct features in the shaping of CP.

**Summary Of The Paper:**

The present study proposes a machine learning framework for predicting cognitive performance (CP), which incorporates a wide range of physiological, personal, behavioral and demographic features. Authors also applied methods such as SHAP and DiCE to get more insights into the contribution of distinct tested features (and adjustments) at both the global and individual levels in shaping CP.

**Weaknesses:**

1) Ethical reference approval for the current study is missing in the manuscript

2) The authors did not disclose the percentage of missing physiological data obtained from the smartwatch comparative to the total amount of data collected, which will allow readers to get an estimate of the imputation work required to complete the dataset.

---

### Official Review · Reviewer_ZBkc · 2025-07-11
**Incremental Contribution on CP Prediction**

**Confidence:** 4
**Clarity Of Writing:** great
**Clinical Significance:** good
**Methodological Novelty:** fair
**Overall Rating:** 4
**Final Rating:** 6

**Experiments And Results:**

good

**Questions For The Authors:**

I appreciate that you note causal inference will be explored in future work. Could you clarify how the current counterfactual recommendations should be interpreted in the absence of causal validation? For example, are they purely associational, or is there evidence to suggest they can reliably guide interventions?

**Strengths:**

The study integrates diverse physiological, personal, behavioral, and demographic factors for CP prediction, moving beyond isolated analyses. Using both real-life oral presentations and standardized assessments strengthens generalizability. The application of XAI tools like SHAP and DiCE adds interpretability and supports personalized interventions. Comprehensive preprocessing, including bias removal, and the use of Leave-One-Out Cross-Validation further support the reliability of the findings.

**Summary Of The Paper:**

Summary Of The Paper
This study introduces an explainable regression framework for predicting cognitive performance (CP) using physiological, lifestyle, and demographic data. The authors utilize machine learning models like XGBoost and Linear Generalized Additive Models (GAM) to capture non-linear and individual-specific relationships within multimodal datasets. The framework incorporates SHapley Additive Explanations (SHAP) for global and local feature attribution to understand how factors such as physical activity, respiration patterns, and heart rate variability influence cognitive outcomes. Additionally, Diverse Counterfactual Explanations (DiCE) are employed to generate personalized recommendations for improving predicted CP. The proposed framework is validated on two datasets: one involving real-life oral presentations and another based on standardized computerized cognitive assessments, showing significant improvements over baseline prediction.

**Weaknesses:**

The main weakness of this study is that it has very small sample sizes (only 30 and 14 participants for the two datasets) which limits generalizability and robustness. Also, while the study integrates many factors, it does not yet include causal inference or longitudinal follow-up — so the recommendations from the counterfactuals may not truly reflect cause–effect relationships. Finally, the data depends heavily on wearable devices and self-reports, which can introduce measurement errors and bias.

---

### Official Review · Reviewer_uNVZ · 2025-07-21
**Towards a Machine Learning Model for Cognitive Performance Prediction**

**Confidence:** 2
**Clarity Of Writing:** good
**Clinical Significance:** good
**Methodological Novelty:** good
**Overall Rating:** 7

**Experiments And Results:**

good

**Questions For The Authors:**

1. Could you elaborate on the specific challenges you faced in integrating such diverse data types (physiological, lifestyle, demographic) and how your framework addresses them?
2. Could you provide more detail on the specific metrics used by the experts for evaluation and how the Bayesian framework quantifies and corrects for evaluator subjectivity?
3. How did you determine the optimal size of this moving window, and what was the impact of this preprocessing step on the model's performance?

**Strengths:**

This study is notable for its integration of a wide range of data, including physiological, lifestyle, and demographic factors. Previous research often focused on isolated correlations.
This paper addresses a gap in the existing literature by presenting a comprehensive machine learning framework that integrates various factors to predict cognitive performance simultaneously.
A significant strength is its ability to provide interpretability and actionable insights into the model's predictions, which is crucial for translating predictions into personalized recommendations.
The framework was validated using two distinct datasets which enhances the robustness of the findings across different contexts.

**Summary Of The Paper:**

This study introduces an explainable machine learning framework for predicting cognitive performance using physiological, lifestyle, and demographic data. It utilizes XGBoost and Linear Generalized Additive Models (GAM) to capture complex relationships and employs SHAP for feature attribution and DiCE for personalized recommendations.

**Weaknesses:**

Both datasets are relatively small.
The study does not fully explore the temporal dynamics of cognitive performance.